# Pros and Cons of CAD/CAM Technology for Infection Prevention in Dental Settings during COVID-19 Outbreak

**DOI:** 10.3390/s22010049

**Published:** 2021-12-22

**Authors:** Livia Barenghi, Alberto Barenghi, Umberto Garagiola, Alberto Di Blasio, Aldo Bruno Giannì, Francesco Spadari

**Affiliations:** 1Department of Biomedical, Surgical and Dental Sciences, University of Milano, 20122 Milan, Italy; umberto.garagiola@unimi.it (U.G.); aldo.gianni@unimi.it (A.B.G.); francesco.spadari@unimi.it (F.S.); 2Department of Medicine and Surgery, Centro di Odontoiatria, Parma University, 43126 Parma, Italy; alberto.barenghi@unimi.it (A.B.); alberto.diblasio@unipr.it (A.D.B.)

**Keywords:** COVID-19, SARS-CoV-2, dentistry, guidelines, reconditioning, aerosol, fomite, safety, intraoral scanning, digital technology

## Abstract

The purpose of this commentary is to update the evidence reported in our previous review on the advantages and limitations of computer-aided design/computer-aided manufacturing technology in the promotion of dental business, as well as to guarantee patient and occupational safety. The COVID-19 pandemic led to an unprecedented focus on infection prevention; however, waves of COVID-19 follow one another, asymptomatic cases are nearly impossible to identify by triage in a dental setting, and the effectiveness of long-lasting immune protection through vaccination remains largely unknown. Different national laws and international guidelines (mainly USA-CDC, ECDC) have often brought about dissimilar awareness and operational choices, and in general, there has been very limited attention to this technology. Here, we discuss its advantages and limitations in light of: (a) presence of SARS-CoV-2 in the oral cavity, saliva, and dental biofilm and activation of dormant microbial infections; (b) the prevention of SARS-CoV-2 transmission by aerosol and fomite contamination; (c) the detection of various oral manifestations of COVID-19; (d) specific information for the reprocessing of the scanner tip and the ward from the manufacturers.

## 1. Introduction

The COVID-19 pandemic is pushing digital technology (DT) towards becoming one of the most crucial and irreplaceable technologies in dentistry, due to its ability to provide serenity regarding safety, increase workflow efficiency, and likely increase profit as well [1,2,3]. In an era of infectious diseases, dental patient safety must always be guaranteed when using all forms of dental technology [4,5]. It is well known that CAD/CAM technology has major advantages that go beyond infection prevention, and which are widely discussed in other papers surrounding this issue.

Nevertheless, the knowledge of infection prevention and its failures is pivotal. Our perspective is to help to create the “correct picture” for dental patient safety during COVID-19. Our purpose is to update the evidence reported in our previous review, on account of the significant changes in dentistry caused by the COVID-19 pandemic [6].

Before a rational approach relating to infection prevention, we report the current knowledge on the presence of SARS-CoV-2 in the oral cavity. SARS-CoV-2 resides in the epithelium of the oral cavity, tongue, throat, nose, and salivary ducts, and is present in saliva and crevicular fluid, even in asymptomatic and mildly symptomatic cases of COVID-19 [7,8]. Immunohistochemical analyses showed that the main receptors (ACE2, TMPRSS2, and Furin) for SARS-CoV-2 are expressed in the stratified squamous epithelium of the gingiva [9]. In addition, it is not trivial for oral-implant surgery, that: (a) gene expression of receptors (ACE2, TMPRSS2) has been shown in human bone cells and osteoblasts [10]; and (b) trabecular bone loss and “full of life” osteoclasts are present in COVID-19 mouse models, even in relation to mild and asymptomatic infection [11]. It is important to note that salivary SARS-CoV-2 was detected, in unstimulated saliva, in 23% of asymptomatic, 28% of presymptomatic, and 60% of postsymptomatic people [12]. Unfortunately, asymptomatic cases are nearly impossible to identify by triage in a dental setting. SARS-CoV-2 causes or influences different oral manifestations, with multiple clinical aspects and oral microbiome derangement [13,14]. Recently, it has been reported that dental biofilm also harbours SARS-CoV-2 RNA from symptomatic COVID-19 patients [15,16]. Added to that, periodontal tissues are a target of SARS-CoV-2 [9,17,18]. Furthermore, periodontal disease and necrotising periodontal disease are associated with a high risk of severe COVID-19 [19,20]. Currently, we do not know SARS-CoV-2 viability associated with dental biofilm, but if present, the virus could evade the oral-mucosal immune response and is in a unique environment for spreading to gingival capillaries, thus sustaining the oral-vascular-pulmonary route of SARS-CoV-2 infection [21]. Microbial contamination of denture surfaces can elicit localised intraoral mucosal infections and has also been implicated in the aetiology of aspiration pneumonia in dependent older adults [22].

In addition, the role of SARS-CoV-2 in the development of oral mucosal lesions (OML) is unclear, as is whether these lesions are true manifestations of the disease, or a secondary phenomenon due to other factors [23]. Many causes should contribute to OML: lysis of infected cells having ACE2 receptor, the immune and vascular disorder, thrombocytopenia, reduction in host antiviral mechanisms, other opportunist infections, medication-related effects (corticosteroids), and multisystem involvement [24,25,26]. OML are different in appearance, size, location, and quantity, and are more common on the tongue, labial mucosa, and palate. The reported oral lesions or clinical consequences are: (a) irregular ulcers, erosions, bullae, vesicles, pustules, or a fissured or depapillated tongue, macules, papules, plaques, pigmentation, whitish areas, haemorrhagic crusts, necrosis, small blisters, petechiae; (b) swelling, erythema, xerostomia and spontaneous bleeding, desquamative gingivitis; (c)in hospitalised children with confirmed multisystem inflammatory syndrome in children (MIS-C): aphthous-like lesions, red or swollen lips, a strawberry tongue [27]; (d) COVID-19 tongue [28]. Proposed mechanisms of different OML have been reported recently by Farid [29]. They can be present before or at the onset of respiratory symptoms in mild cases, or after 7–24 days after the onset of symptoms in hospitalised cases. During SARS-CoV-2 infection, OML are similar to those of chronic autoimmune and/or inflammatory conditions, such as lupus erythematosus and those of patients that are undertaking long-term immunomodulatory therapy (using corticosteroids), or with Sjogren syndrome, or pemphigus. Patients with OML had symptoms such as burning sensations, pain, and pruritus in 68% of the cases [26]. Some of the oral lesions (pigmentation, reddish lips, erosive lesions on tongue) have been observed in a patient with a history of asymptomatic COVID-19 or as a persistent symptom of long COVID-19 without other signs of COVID-19 and a negative COVID-19 test [30,31]. Unfortunately, the use of a specific compound in prophylactic pre-procedural mouth rinses to reduce mucosal and salivary SARS-CoV-2 is not supported by clear evidence [12,32,33]. In addition, oral ulcers can be caused by the extensive use of topical antiseptic oral applications (such as hydrogen-peroxide-based mouth rinse solutions), recommended to reduce the oral SARS-CoV-2 load, [34,35,36]. The judicious use of hydrogen peroxide is recommended to limit the damage to softer oral tissues in geriatric dental patients. 

Unfortunately, it is well known that the first wave of COVID-19 caught dentists largely unprepared in terms of facing an airborne pandemic of this nature. This led to unprecedented focus on infection control and prevention (ICP). The first recommendations were *Ad Interim* guidelines by the Centre for Disease Control and Prevention (CDC), followed by those of the national and professional boards [34,37,38,39,40]. During the first wave, recommendations for non-elective dental care were not based on solid scientific evidence, but rather, for the most part, followed experts’ opinion-based studies on ICP in dentistry. In part, these guidelines were related to the short outbreak of SARS-CoV-1, and were mainly made for hospital settings. Subsequently, four other waves of COVID-19 followed one another in Europe and all around the world [41]. The COVID-19 occupational risk for dental health care workers (DHCW) and dental patients is a burning issue, based on largely unknown evidence [42,43,44]. Recent evidence shows that the rate of SARS-CoV-2 infection is the highest in DHCW, at 26%, while in health care workers it is at 14.5%. These figures are, respectively, about 5- and 3-fold higher than in the local population [44]. Even so, it is not reassuring that data from a 6-month longitudinal study from June to November 2020 shows that only about 60% of dentists always wore a filtering facepiece (FFP) respirator and eye protection when performing aerosol-generating procedures (AGP). Additionally, the extended use (no change for every patient) of respiratory protection has been adopted by 75% of dentists [43].

During Phase 1 of COVID-19 only treatment for dental emergencies was recommended. Then, during Phase 2, elective dental procedures also resumed and in the meantime many populations went through broad anti-COVID-19 vaccination. Safety for DHCW and dental patients was vital throughout. In general, the information required to sustain evidence-based recommendations for dentistry is slowly coming to the surface [42,45]. It is clear that the concept of routine normality in dentistry, as in the pre-COVID-19 era, will never be the same [46]. Only DHCW that can adapt to a frequently changing scenario will be able to move forward successfully and safely. To the best of our knowledge, specific recommendations on computer-aided design/computer-aided manufacturing (CAD/CAM) technology use during COVID-19 are lacking in national or international guidelines and recommendations; few expert papers were published containing scarce recommendations [47,48]. 

Here, we discuss the advantages and limitations of DT in sustaining dental business and in safeguarding patient and occupational safety. The essential adaptations [6] to the COVID-19 pandemic concern four areas: (a) presence of SARS-CoV-2 in the oral cavity, saliva, and dental biofilm and activation of dormant microbial infections; (b) the prevention of SARS-CoV-2 transmission by aerosol and fomite contamination; (c) the detection of various oral manifestations of COVID-19; (d) specific information or manufacturer information for users (MIFU) for the reprocessing of the scanner tip and the scanner ward. 

## 2. Materials and Methods

### Information Sources and Search Strategy

This paper is a narrative review regarding the recommendations and guidelines for infection prevention using CAD/CAM technology in dentistry during the COVID-19 pandemic. An additional goal is to disclose the hidden side of SARS-CoV-2 prevention during its use. Without language restrictions, the electronic literature search was conducted via the PubMed and Google Scholar databases (from November 2018 up to and including September 2021) using the following key indexing terms: (a) CAD/CAM dentistry; (b) cross-infection control; (c) infection prevention; (d) disinfection; (e) reconditioning; (f) semi critical items; (g) critical items; (h) cast; (i) digital model; (j) digital impression scanner; (k) dental impression; (l) guidelines; (m) safety precautions; (n) dental laboratory; (o) occupational health; (p) bacterial adhesion; (q) microbial contamination; (r) biofilm; (s) COVID-19; (t) SARS-CoV-2. Then, we used the combinations of key indexing terms as follows: (a) or (j) or (k) with (s); (a) or (j) or (k) with (t); (l) or (m) or (o) with (s); (l) or (m) or (o) with (t); (b) with (s); (c) with (s); (b) with (t); (c) with (t); (o) with (s); (o) with (t); (q) with (s); (q) with (t); (h) with (p); and (a) with all other single key indexing terms. In addition, manual searches were carried out in the MDPI Journal database (up to 18 September 2021) using the following key indexing terms: (a) CAD/CAM (n° = 274); (b) CAD/CAM dentistry (n° = 33), COVID-19 dentistry (n° = 76); however, only one of these takes into account cross-infection or infection prevention according to our specific topic [49] or reported relevant and specific recommendations for CAD/CAM technology concerning COVID-19 prevention. Since the beginning of the COVID-19 outbreak, we checked the recommendations and guidelines for infection prevention by international boards (WHO; CDC USA; ECDC) and national boards (UK, Canada, EU nations) for healthcare and dental care in outpatient settings. Subsequently, bibliographic material from the papers, guidelines, and recommendations has been used in order to find other or older appropriate sources in relation to specific topics and operative problems. A total of 136 papers and links were found suitable for inclusion in this paper. Only a few documents, mainly MIFU [50,51,52,53,54,55,56], do not have a DOI or PubMed classification, but the available internet link and date accessed have been added. In this review, the inclusion criteria were all documents, validated by at least three authors for each specific knowledge area, which included unambiguous information on CAD/CAM technology and relevant information for COVID-19 prevention in dental settings.

## 3. Results and Discussion

Dental care is a priority because it is well known that the lack of or delay in dental care can have adverse consequences on oral and non–oral health of the population, especially in vulnerable people. During the COVID-19 pandemic, low patient motivation for oral hygiene and dental care is dangerous for oral health, for the long-term success of all dental care, from caries prevention to dental implantology or orthodontia, and, no less crucially, for the economic future of the dental office [57,58,59,60]. The detailed list of clinically, extremely vulnerable groups for COVID-19 has been reported elsewhere, and includes mainly immune-compromised people [49,61,62]. That is to say, vulnerable persons are those in a particular physiological state regarding their immunity (such as pregnant women, youths, and adolescents) or vaccination (non-responder or immunocompromised people), or people receiving systemic pharmacotherapy with angiotensin-converting enzyme (ACE) inhibitor or long-term immunomodulatory therapy (corticosteroids). 

The COVID-19 pandemic also greatly affected dental patients. They are more vulnerable to COVID-19 due to their medical, physical, intellectual, and learning disabilities, more stressed, more phobic, more aggressive, and more alert to dental costs [63,64,65,66]. This is in line with the fact that dental patients always require tangible, operational safety for the prevention of any infections, and the fear of contracting SARS-CoV-2 and anxiety [67] (p. 2). In addition, the following problems were encountered: patients’ pain in relation to dental emergencies, pain/loss of function due to broken prostheses, an increase in discomfort and low motivation because of missed appointments, increased treatment time for orthodontic work, and delayed oral hygiene and dental implant care [64]. During the pre-COVID-19 period, DT and CAD/CAM technology was developed mainly for highly specialist dental care (prosthesis, implantology, and orthodontia) [68,69]. There was a clear increase in scientific publications related to digital implant dentistry, and most key indicators demonstrated an upward trend [70].

### 3.1. Main Advantages of CAD/CAM Technology for COVID-19 Prevention in Dentistry

During the COVID-19 pandemic, the distinctive advantages of DT are the reduction of close contact with patients and DHCWs (within 6 feet), and limited transmission through respiratory droplets and AGP. These features are very important for COVID-19 prevention, and in particular, for special care dentistry, dental care on non-responder patients to COVID-19 vaccinations, and face to face care in patients affected by long-COVID-19 syndromes [71]. Long-COVID-19 syndromes are largely unknown, but at least two features are relevant to dentistry: breathlessness (25% of cases) and dysfunction of the oral microbiome [13,71].

In general, the advantage of digital over conventional workflow is to perform digital simulation and virtual planning (mainly for surgery and orthodontia) and then several treatment steps are possible with software and with no need for human contact. An easier way to contain the spread of infectious agents is by decreasing the number of steps and by lowering the number of procedures which likely generate aerosol and environmental contamination. This includes the digital impression procedures which reduce the gag reflex, the need for the prone position to cough, and which are more pleasant, mainly for elderly patients [61,69] who suffer from respiratory diseases, including long COVID-19, or chronic or acute oral mucosa diseases. Intraoral scanners (IOS) result in significantly decreased discomfort in the mouth opening (e.g., temporomandibular joint disorders and implant dentistry), vomiting, and breathing difficulty. Additionally, it rarely affects periodontal sensitivity because it does not require excessive or a long period of mouth opening as compared with conventional impressions [72]. Furthermore, analogic impression procedures require handling multiple items and they fail certain steps in infection prevention [6] (pp. 3–8). 

During the COVID-19 pandemic, the CAD/CAM approach supports the change from a repair model of care to a wellness-based model and a person-centred approach. Using real-time imaging, it allows for individualised feedback on one’s own oral health, care of patients, patient choices, and possible outcomes. It also allows for the preparation or modification of new or broken dental devices (for dentures, bridges, crowns, veneers, and orthodontic appliances or procedures), guides for a precise dental implant insertion with a reduced number of appointments, shorter chair-side time, as well as higher precision and accuracy, when compared with conventional methods [3,49,57,61,68,69,73,74,75]. Even dental plaque examination by IOS is an advantageous new approach for dental hygiene instruction and evaluation (by binarization of the 3D images); in addition to preventing infection, the evaluation and recording can be performed by one person while the direct method requires two people (potentially exposed to SARS-CoV-2): an evaluator and a recorder [76].

As a final point, DT is very important because of the increased time needed by dental assistants for decontamination procedures between patients during the COVID-19 pandemic and DHCW’s need to reduce the frequency or stay of patients in the dental office. More efficiency and cost-effectiveness seem to be promising for the reduction of dental costs for patients and improving oral health and wellness to cover the population’s need [2,3,69]. Nevertheless, the development of the professional harmonization and ergonomics of the “human-technical complex dental office system” is necessary in light of the newly emerging infectious agents, the health service disruption, the expected aggravation of the prevalence of infectious agents (*Mycobacterium tuberculosis*, drug-resistant infective agents), well known in dentistry, and the supposed endemic fate of COVID-19 [13,77,78].

#### Additional Advantages of Digital Scanning to Avoid Compression of Mucosal Tissues and to Detect OML

Concerning the unease of OML, it is well known that digital scanning does not cause compression of mucosal tissues and that it is more comfortable for patients compared to traditional impressions. In addition, it allows for the immediate availability of enlarged images on a PC screen, with a reduced use of intraoral mirrors. Obviously, the analogic impression is probably more contaminated on the corresponding site of OML because of the many microbes involved in them (opportunistic fungal infections, HSV-1, HSV-2, varicella zoster virus, Coxsackie viruses, HPV) and generally plaque-associated infective agents. Concurrent oral exfoliative cytology and intraoral scanning would be very interesting, easy to use, and advantageous approaches to screening asymptomatic patients and drug therapy efficacy [79].

### 3.2. Market Perspective and Dentist Expectations during COVID-19 Pandemics

#### 3.2.1. Dental CAD/CAM Market Size, Share & Forecast, 2021–2027

COVID-19 has had a detrimental effect on the dental CAD/CAM market for different reasons: government-imposed restrictions on elective dental care, social isolation, and patient fear of infection. The global market exhibited a decline of −4.8% in 2020 compared to the average year-on-year growth during 2017–2020 [80]. There has been an increase in demand for telemedicine and the adoption of a fully digital prosthetic workflow that incorporates CAD/CAM technology. The sudden growth is expected because dentists need to increase safety, profit, and workflow efficiency, and these can be more readily met using DT in-office systems. Global dental CAD/CAM market size was valued at more than USD 1.5 billion in 2020 and is estimated to grow at a CAGR of 5.2–11.5% between 2021 and 2027 [81]. The Americas are expected to dominate the global dental CAD/CAM market in the coming years, and Europe is the second-largest market. Additionally, Europe is a leader in the use of advanced technology and upgrading of their equipment, including intraoral scanners. An increasing number of practicing dentists and a growing demand for chair-side CAD/CAM systems are expected to boost the market value during the forecasted period. The shortage of customised PPE during COVID-19 will increase the development of 3D printing used for the production of medical protective equipment [82], masks, shields, or connectors to breathing devices [83]. 

#### 3.2.2. Dentist Perspectives

The use of DT, also requiring fewer intermediate checks, is expected to be money-spinning for busy dental practices and dental technician laboratories; in addition, a positive effect on the reputation of the dental practice ensues [84]. CAD/CAM (chair-side) has been reported to have higher economic benefits than the scanners used only for digital impressions [85]. A study published recently showed that digital impressions are more efficient and cost-effective than standard impressions, and implementation costs can be offset within the first year of dental practice [2]. A significant reduction in working time (~27%) and number of appointments (~17%) has been reported when a fully digital workflow was implemented, compared to the conventional workflow [3]. This is important for elderly patients, because of the need to reduce risk by having the fewest appointments possible [86]. Until now, there was no data on long-term and overall economic profits after CAD/CAM technology adoption, including learning curve cost, time to solve new difficulties and problems, reduction of the use of impression disinfectants and limited infectious waste disposal, low transport costs, and the duration of the aesthetic and functional guarantee of the customised medical device (MD) with new materials. 

The scanning by a dental assistant has been proposed to further reduce cost [87] and provide a more active role of the dental assistant in patients’ treatment. Supported by Italian laws, we are absolutely against this strategy: we need highly trained DHCW for scanning and dental assistants must be primarily committed to the careful ICP and communication policy. Dental nurses already have an exceptionally laborious and stressful job during the COVID-19 pandemic.

### 3.3. The Trouble of Miscellaneous Laws, Guidelines, and Recommendations

In general, even in the case of the similar legislation and strong connection among the EU countries, a recent review reported the heterogeneous and slightly standardised versions of national guidelines of EU countries for the management of phase I and II [39]. The majority of countries recommended both elective and urgent treatments, but the definition of urgent and elective treatments was not consistent or always provided. 

The first ECDC recommendation was only published by European Centre for Disease Prevention and Control (ECDC) on October 2020, has not been updated yet [88], follows the recommendations of the international organizations [32,34,37], and did not take into account impression disinfection. 

During deferrable and non-deferrable dental emergencies for suspected or confirmed cases of COVID-19, the main indications are to minimise AGP (including reduced use of the air/water syringe), adopt four-handed work and use the rubber dam, use proper PPE, high-volume intraoral suction, and environmental disinfection. It is a requirement to schedule appointments in a way that maintains satisfactory indoor air quality, and to apply stringent environmental and careful MD reconditioning [32]. 

There are probably benefits to using these main interventions, but the evidence to support them is weak and mainly based on effects on bacteria, rather than viruses or respiratory diseases [89].Concerning AGP, there is strong agreement in the literature to include the use of air/water syringes and the use of high-powered lasers among the high-risk procedures, while the use of air-water syringe, with water only, is classed among the lower-risk procedures [90]. It is well known that the use of air/water syringe is a critical step before impression taking. Dental impressions, trays, occlusal records, prostheses, or appliances can be contaminated with bacteria, viruses, and fungi. Impression disinfection by conventional procedures is not considered among many ICP behaviours during COVID-19 phase II, even in recent papers, when elective (deferrable) dental care was permitted [6,91,92,93,94]. It is risky that, concurrent with limited knowledge about fundamental aspects of disinfection protocols, analogic impression disinfection was not at all considered [95].

To our best knowledge on traditional practices and DT, insufficient notes are available from laws on health safety and guidelines during COVID-19 pandemics [39,96]. Concerning DT, specific procedures for the limitation of SARS-CoV-2 contamination have not been considered, even in the case of prosthetic or orthodontic emergencies. The Organization for Safety, Asepsis and Prevention (OSAP), the only non-profit membership association for oral health care professionals that focuses exclusively on infection prevention and patient and provider safety, also seems uninterested [97]. Further to that, despite the advantages and high interest in CAD/CAM technology in dentistry during COVID-19, gold standard guidelines for infection prevention have not yet paid attention to it [32,34,37,45,98]. The lack of interest is caused by: High cost of CAD/CAM devices and its diffusion mainly in high-income regions;Low affordability in underdeveloped regions;No or low perceived advantages over traditional procedures;Psychological resistance toward the shift from analog to digital dentistry in practice;Need for highly trained personnel;Not lucrative in many business models [84].

#### 3.3.1. Other Recommendations for Prosthodontics and Their Limits

In general, specific recommendations have been reported by other boards for prosthodontics practice, with mainly traditional procedures during lockdown [48,61]. Elective dental care and non-emergent dental treatment have to be deferred for a patient with suspected or confirmed SARS-CoV-2 infection. Nevertheless, elective care, including prosthetic care and surgery, should be provided unintentionally to asymptomatic or presymptomatic patients. Different urgent prosthodontic procedures, caused by pain/loss of function due to broken prostheses, have been reported mainly for older patients, and require certain chair-side protocols. Obviously, the use of a rubber dam for containment and protection from oral fluids (saliva, crevicular fluid), often contaminated by occult blood, is very limited (onlay scanning), and it is largely not applicable during tooth preparations, with lower gingival margins, impression procedures, and occlusal recording. The advantage when using erbium laser seems promising during prosthodontic treatments in the light of reduced production of aerosol when compared to conventional rotatory tools and plume biohazard containment by HVE [99]. Due to the lack of specific guidelines for impression disinfection and dental laboratories on infection control during COVID-19, some experts published some recommendations [61,100,101]. Some recommendations for analogic impression and MD disinfection for geriatric dental practice are not based on specific evidence surrounding SARS-CoV-2 [86,98].

Higher rates of bacterial colonization, especially with *Streptococcus* species and *Klebsiella pneumonia*, were detected in oral samples and on removable dental prostheses after COVID-19 infection [102]. There are no data on the contamination of dental impression and the virus survival, but the saliva contamination, comparable to nasopharyngeal specimens, suggests the possibility of viral bioburden in addition to bacterial and yeast ones [103]. Traditional procedures are expected to be at higher risk for impression and environmental contamination by SARS-CoV-2 and blood (by compression of inflamed gingival tissues) compared to digital procedures. It is important to note that during the lockdown in Wuhan, some infected DHCWs worked in the prosthodontics department [104].

#### 3.3.2. Recent Indication from CDC for Environmental Infection Controls during COVID-19 Pandemics

The recent recommendations are [32,65,105]:The use of specific types of disinfectants and concentration (sodium hypochlorite, hydrogen peroxide, and povidone) has been cancelled since it is not sustained by sound evidence, and rather mainly by data from experiments in vitro on SARS-CoV-2;All non-disposable medical equipment used for a patient should be cleaned and disinfected according to MIFU;Routine cleaning and disinfection with EPA-registered, hospital-grade disinfectants with appropriate claims (efficacy, kill time, compatibility, safety) for frequently touched clinical surfaces are appropriate for SARS-CoV-2, also in areas where AGP are performed;Refer to list N on the EPA website for EPA-registered disinfectants that kill SARS-CoV-2.

### 3.4. From Recommendations to Practice for Infection Prevention during COVID-19 Pandemic

Some evidence on unpleasant microbial transmission in patients and DHCW cannot be forgotten [4,5,43,44,104]. It is widely accepted that patient and DHCW safety relies on understanding the types and causes of errors and possible causes of infectious injury and on the planning of recovery actions. In the further sections of this paper, our aim is to examine the difficulties and pitfalls that may occur during the intraoral scanning, and the reconditioning of ward, scanner, and scanner tips. 

#### 3.4.1. Focus on Challenges Regarding Analogic Impression

In general, the main recommendations are related to the infective danger during the carrying out of dental impressions by conventional procedures, but without evidence-based contamination by SARS-CoV-2 and specific notes on disinfectants for impression, cast, scanner, tips, and hardware for CAD/CAM [47]. We would like to note that both during the traditional and digital carrying out of dental impression, the dental staff are at risk because of close contact with patient droplets and aerosol, but they experience different exposure to oral fluid and AGP procedures (including the use of syringe air/water). Different exposure times are related to manual mixing and setting times of the impression material compared to scanner speed. Procedures on dental prostheses, impressions, orthodontic appliances, and other prosthodontic materials (e.g., occlusal rims, temporary prostheses, bite registrations) had been indicated in previous CDC guidelines [105]; items should be thoroughly cleaned (i.e., blood and bioburden removed) and disinfected with an EPA-registered hospital disinfectant with a tuberculocidal claim. Tuberculocidal claim of a disinfectant should be rationally sufficient to inactivate SARS-CoV-2, an enveloped virus. In general, analogic impression and MD disinfection for geriatric dental practice are not based on specific evidence for SARS-CoV-2 kill [69,98].

Nevertheless, the immersion techniques are more effective than spray in reaching all the surfaces and will be safer in the presence of an airborne disease, but they are not recommended for alginate and polyether impressions. Furthermore, it is expected that environmental factors (hydrophobic vs. hydrophilic surface) affect the viability of SARS-CoV-2, with the longest life span at low temperatures, high humidity, and dirty conditions [106,107,108]; these are peculiar features of storage of alginate impressions. The average survival time on plastic or steel material, used for impression tray, is approximately 72 h in vitro. Traditional procedures need a written prescription (also known as a work of order or requisition) in two copies from the dentist. This is a crucial step because SARS-CoV-2 is shown to be vital and infectious on paper for 24–36h. It is well known that prescriptions can be frequently contaminated by visual imperceptible blood, saliva, and airborne contamination. 

Due to the lack of specific guidelines for impression disinfection and dental laboratories on infection control during COVID-19, some experts published some recommendations [34,61,100,101]. The recommendations are only for traditional procedures with more attention to the preservation of dimensional stability and impression accuracy than disinfectant efficacy. Studies have been proven for enveloped virus (EN14476—dirty conditions), but rarely for SARS-CoV-2 [25]. No attention is given by MIFU to spike protein’s different stability (at neutral-acidic pH) in relation to disinfectant pH (pH range: 4–11.5) for impression and devices [109]. The COVID-19 pandemic was not sufficient to improve knowledge and poor compliance concerning infection prevention among dental technicians [110]. 

#### 3.4.2. SARS-CoV-2 and Factors Influencing Intraoral Scanning and Digital Practice

In simulated conditions, it is well known that saliva can lead to loss of accuracy of IOS, with variable accuracy and values that are not clinically accepted [111]. Obviously, the rubber dam allows for isolation of the dental surfaces, but only in extra gingival preparations. We previously reported the factors influencing intraoral scanning for digital imaging [6] (p. 9). Here, the advantages and disadvantages of CAD/CAM technology compared to analogical procedure have been enlarged in Table 1. 

A clean and dry field (free from oral fluids and blood) and tissue retraction management are requested in order to obtain an acceptable image by scanning. Nevertheless, we would like to underline that periodontal tissues are the target of SARS-CoV-2. The possible interference of oral fluids or its components will be taken into consideration when using intraoral scanners for dynamic depth technology, which provide scanning of up to 20 mm in depth [112]. In the future, some interference is expected to be solved by incorporating the scanning process with tomography or ultrasound-based imaging technology. The latest technology uses new generation probes and ultrasound waves to assess, image, and locate the upper border of the periodontal ligament and its variation [113]. 

It is unknown if arterial and venous thromboembolism and higher blood clot risk caused by insoluble micro clots during COVID-19 and in vaccinated patients can influence digital impression acquisition on deeper surfaces of a stump or implant [114]. Caution is needed because of the links between vascular damage, periodontal disease, and SARS-CoV-2 presence in oral tissues. 

#### 3.4.3. Focus on Reconditioning of Digital Scanner and CAD/CAM Materials

##### Reconditioning of Digital Scanner

We have to strictly adhere to updated MIFU and visually inspect any accessory for signs of deterioration after reconditioning and before scanning. Improper sterilization may cause small cracks on the scanner mirror. Then, acquired images would contain noise signals which may slow the scanning speed and require replacement of the scanner tip. According to CDC classifications, most intraoral scanners are composed of non-critical (the scanning unit (wand), cradle, touch screen) and semi-critical surfaces (tips which cover the sensor, including any mirrors), because only the latter come into contact with mucous membranes. Nowadays, the majority of scanner tips are removable and autoclavable, according to CDC guidelines [105,115].

The presence of SARS-CoV-2 in the oral cavity, also in asymptomatic patients, makes the contamination of scanner tips possible, especially in areas restricted to scanner access. The potential causes are: (a) inability of the patient to sit still; (b) large head of the tip; (c) no ergonomic design/wand size; (d) no availability of specific tips for adults, child, neonates; (e) scanning teeth lingual surfaces (tongue movements) or vestibular molar surfaces (near the parotid salivary duct). In fact, SARS-CoV-2 is located mainly on the tongue and in the salivary duct (see Introduction). Nevertheless, the greater the scan distance, the lower the accuracy, for all the scanners [116]. Reasonably, the parallel confocal technology need not require a certain distance for focusing, thus ensuring accurate images irrespective of whether the scanner tip is in contact with the teeth when the oral cavity is scanned [117]. Infection prevention should take advantage of mesh quality and scanning technology (photographic or video-based scanning) not influenced by lighting conditions [118]. However, the main disadvantages of intraoral scanning are breathing and saliva secretion, which cause deviations, interfering with the applicability and accuracy of the optical impression [6] (p. 9) [111]. 

During COVID-19 outbreaks, mainly to prevent air contamination, we prefer to use:Powder-free intraoral scanner system because fine and ultrafine particles contained in scanning spray may be harmful to the respiratory tract and may cause possible air contamination [6] (p. 9).A scanner tip heated with a heat wire and not with air flux to prevent fogging, even if it requires a waiting time to allow the scanner tips to be sufficiently heated compared to air.

##### Current MIFU for Dental Scanner

Here, we enlarge some current MIFU for CAD/CAM technology to prevent SARS-CoV-2 transmission (Table 2) [6] (pp. 6–8). Low hand touching is a challenge to reduce SARS-CoV-2 transmission, even if its transmission by aerosol and fomites is largely unknown in dental settings [32,119,120,121]. Some current advantages of CAD/CAM include the inclusion of touch screens and wireless technology and e-data storage, which are quick, secure, and easy to share, without the need for “paper, pen and hand”. MIFU have been clarified and enlarged during COVID-19, often using YouTube videos and some very clear images of damaged mirrors of the scanner [50,51,52,53,54,55,56]. That said, MIFU were not specifically related to COVID-19 prevention and mainly focused on USA recommendations rather than EU directives. For safe reuse, MIFUs include procedures (mainly using non-abrasive impregnated wipes or immersion when possible) and specific products for the: (a) cleaning (with soapy water and a soft brush or non-abrasive wipe wetted with an enzymatic detergent solution); (b) intermediate-level disinfection (using approved disinfectant wipes, with specified alcohol amount or alcohol without impurities or alcohol-free); (c) sterilization by steam autoclave or high-level chemical disinfection of the scanner tips; (d) intermediate-level disinfection for ward, cart, and mobile [50,51,52,53,54,55,56]. 

There are two main differences between EU and US recommendations. European DHCW need to select glutaraldehyde-free disinfectants according to European law governing occupational safety, while the use of glutaraldehyde is allowed in USA. These constraints influenced MIFU accordingly [50,56]. In addition, small steam autoclaves for dentistry have been developed according to EN13060. European DHCW focus more on vacuum and steam penetration controls and chemical controls (integrator class 5), rather than on biological tests (spore test). Both sterilization cycles with pre-vacuum autoclave (at 121; at 132 for at least 4′ or 134 °C for 3′ and minimum drying time (20–30′)) are normally allowed, but some cycles (autoclave sterilization cycle times exceeding 10′), or the absence of protection (gauze covering the mirror), reduce the scanner tips’ life and cause mirror staining, which is often permanent [54]. We underline some confusing MIFU concerning: (a) careless (not from clean to dirty area) movements during disinfection with wipes impregnated with forbidden chemicals [56]; (b) approved commercial disinfectants [55]; (c) paper sterilization pouches [50] (p. Italian version); and (d) sterilization time for different cycles using gravity or pre-vacuum autoclave [54]. The scanner tips can be steam sterilised, from 60 times (sterilization at 134 °C) [54], up to 150 times [51,52,53]. 

In addition to the indications reported in Table 2, we summarised that DHCW DO NOT:Apply liquids directly to any surfaces of the systems;Make careless movements, instead always go from cleaner to dirtier areas;Use isopropyl alcohol on the cart touch screen;Use window cleaner or other chemical cleaners on touch screens;Place scanner ward and tips in an ultrasonic cleaner;Disinfect the scanner using cotton, cloth, or tissues soaked with disinfectant, in order to avoid disinfectant inefficacy, deterioration and fraying of fibres, and faded stains on mirrors; andUse disinfectants on any surface containing: phenolics, iodophors, ammonia, chloride, or acetone, otherwise indicated as acids, bases, oxidizing agents, or solvents [55]; doing so will damage the surface coating of the scanner.

We recommend attention to these factors because some disinfectants, sodium-hypochlorite-based (0.1–0.5%) and hydrogen-peroxide-based (0.5%), with a contact time of 1’, were listed as killing SARS-CoV-2 in many recommendations and CDC recommendations until November 2020 [34]. Some mirror or wand deterioration is expected if hypochlorite, povidone or hydrogen-peroxide-based disinfectants have been used. The scanner shell may turn yellow/brown or crack after long use of unapproved products by producers.

#### 3.4.4. Focus on Digital Models

Digital study models offer a reliable alternative to traditional plaster models, which are the gold standard in orthodontia, but with several disadvantages (cost, storage space, degradation, potential loss during transfer). The virtual articulators allow for the simulation of patient-specific data in real time and analyse dynamic and static occlusion without hand touching. In addition, because of the gypsum cast’s high contamination in the traditional procedure (Figure 1a,b), gypsum can be substituted with synthetic materials that are non-porous, allow lower levels of microbial colonization, and that can be easily decontaminated with liquid disinfectants (Figure 1c,d). Surface contamination (identified in RLU with 3M Clean-Trace ATP surface test) can be tens of thousands of times higher on plaster models than those in epoxy resin material in full digital CAD/CAM technology (Barenghi L, et al., data not shown).

### 3.5. The Challenge to Dental Surgery during COVID-19

The wide oral presence of SARS-CoV-2 in mucosal and probably trabecular bone cells and an altered microbiome have been discussed in Section 1. In addition to better guidelines on infection prevention during dental surgery, studies highlighting the effects of the COVID-19 pandemic on regenerative bone surgery and recommendations for patient selection are welcome [122]. Oral surgeries are at risk of *enterococci*, hepatitis virus, and SARS-CoV-19 transmission [4,5,9,11,12,17,18,43]. Preliminary simulation by DT allows a more precise and quick surgical technique, fewer surgical errors, reduced potential contamination of surgical sites, and lower risk caused by respiratory, vascular, and coagulation pathology in cured COVID-19 patients or people with long-COVID-19 syndrome [40,106,121,123]. It is well known that the implant-abutment microgap magnitude is important for the long-term clinical success of dental implants. The CAD/CAM abutments presented smaller microgaps [124]; microgaps (1–9 µ) make the entrance of bacteria into the favourable niche inside the implant difficult. Taking into account causes of adverse events, the prosthetic restorations made using CAD/CAM are not free of defects, as reviewed by Skorulska [125]. Recurrent periodontal diseases have been reported for reinforced glass ceramics and glass-infiltrated alumina. This evidence should be investigated to avoid adverse events mainly in elder patients with long-COVID-19 syndromes and periodontal disease.

### 3.6. Medico-Legal Perspective on Providing Dental Care Using CAD/CAM Technology during COVID-19 Pandemic

In the absence of laws or guidelines specific to CAD/CAM technology, the practitioner is required to deliver care in an up-to-date and evidence-based manner, and to sustain the use of CAD/CAM technology to further reduce the risk of spreading and transmitting COVID-19 compared to traditional procedures [96,126]. Nevertheless, to avoid legal consequences, the dentist needs to record changes to the traditional operating procedures and the reason for them. Otherwise, practitioners are considered to have violated a requirement imposed by laws or guidelines. Digital images of pre-existing and final clinical conditions can support guarantees and speed up procedures (authorization, payment, refunds) for an insurance claim.

#### Certification and MIFU

Control and surveillance of medical devices and their MIFU, from institutions have to be increased for two reasons. Manufacturers must follow national criteria (FDA or EU Regulation n° 745/2017) [127] for reprocessing instructions of MD and must develop validated reprocessing instructions. If the MIFU for reconditioning are not clear, as indicated in Section 3.4.3.2, dental users must achieve clarified MIFU in writing [128]. 

Secondly, dental devices and equipment and their upgrades are expensive because CAD/CAM systems for dentistry involve a complete package of components (scanners, designing software, manufacturing software, and milling machines or 3D printers). As a consequence of the economic struggles and difficulties of the dental practices during the COVID-19 pandemic, dental users are more receptive to cheaper products. The risk is to buy grey market products which do not adhere to industry standards, and therefore can be ineffective, unsuitable, expired, and can put the patients’ health at risk. We recommend the use of proper items with FDA and/or CE mark [127] with particular attention to disinfectants for clinical contact surfaces (scanner and hardware), hand hygiene products, PPE (shield, FFP, and glove), and barrier quality [67] (pp. 81–83) [129,130].

### 3.7. Future Trends for ICP

In addition to the surface disinfection by more compatible chemicals and the use of new barriers (silicone-made for hardware or transparent without any light/laser interference), a future opportunity will be the adoption of innovative materials (polymeric-based copper-oxide-impregnated materials or copper coating or copper-based nanoparticles) for the external surfaces (scanner, keyboard, mouse) having antibacterial and antiviral efficacy to reduce multiple surface-hand-skin transfer [131].

#### Reconditioning of CAD/CAM Materials

Only recently, there has also been attention paid to disinfectant compatibility for dental MD (in particular PMMA prosthesis). Studies have been reporting microbial adhesion (*Candida albicans*) to denture bases. The milled group displayed promising potential for reducing the adherence of *Candida* [69]. Because of the ban on glutaraldehyde to avoid work risk in the EU, chlorhexidine digluconate is the best choice in terms of avoiding changes in roughness and colour changes in CAD/CAM materials [132], compared to conventionally fabricated acrylic resin [133]. The use of povidone iodine or sodium hypochlorite, generically indicated by the CDC during phase I and II, has not been supported by scientific evidence for SARS-CoV-2 prevention and compatibility with acrylic appliances [134]. 

Recently, Dusmukhamedov’s group described the reduced infectious hazard and benefits of a fully digital workflow and the fabrication procedure of complete dentures based on digital impressions of edentulous jaws [135]. An interesting advantage could be the reduced microbial contamination caused by the lack of bonding process errors on a monolithic denture and smoother surface. The roughness of the surface increases the level of biofilm adhesion. Nevertheless, increased fungal adhesion has been observed on oral splints manufactured by 3D printing and milling [136], probably due to insufficient polishing after milling. This may be relevant for medically compromised patients, or patients affected by long COVID-19, or asymptomatic COVID-19 patients, or vulnerable patients, including elderly people: these are prone to infections, and are often on antibiotics, situations which favour antibiotic-resistant pathogens.

## 4. Conclusions

For patient and DHCW safety against infectious and SARS-CoV-2 risk, many structural, electronic, and operational characteristics of scanners and CAD/CAM technology are important (Table 2). The main innovations are expected to be:Voice command to avoid hand touching;Antimicrobial surfaces and antimicrobial-coated barriers, silicone barriers, and optical-medical-grade barriers to avoid surface contamination;Wand tip smaller dimension and speedier scanning to reduce close contact with patients;Scanner accuracy improvement;Duration (increased number of sterilization cycles) of scanner tips;Hardware and disinfectant products made of more ecologic and green materials.In addition, concerning particularly the reconditioning of scanner and scanner tips, we advise the users of CAD/CAM technology:To advocate for the creation of specific guidelines for dentistry;To demand clearer and better-written MIFU for dental scanner and ward;To review risk assessments for SARS-CoV-2 and chemicals (disinfectant and cleaners);To improve functions and alerts for reminding the user to perform cleaning/disinfecting procedure on hardware according to protocol;To identify errors and near misses during mostly manual work when reprocessing IOS and scanner tips.

## Figures and Tables

**Figure 1 sensors-22-00049-f001:**
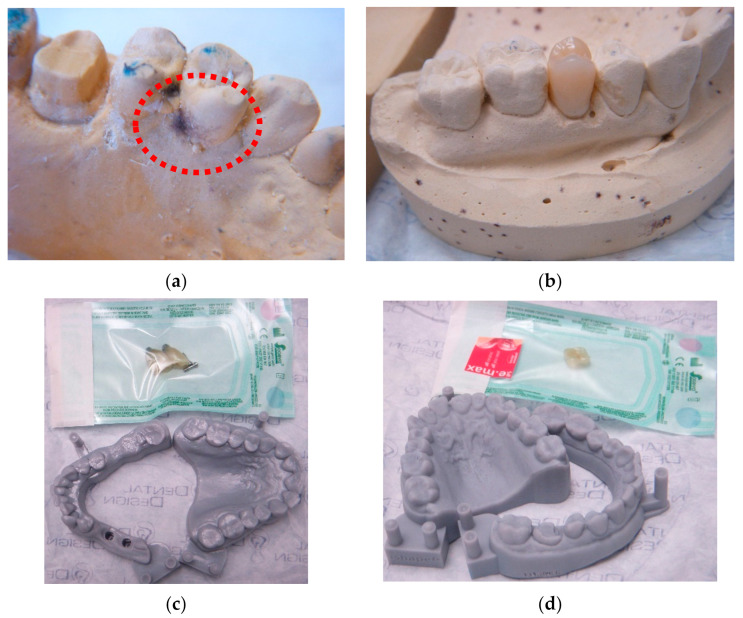
Casts by traditional and digital technology. (**a**): examples of microbial cast contamination (see bluish-black hairy colonies probably from fungus species) using a traditional technique. The casts represent a hazardous reservoir due to DHCP hand touching; (**b**): an example of lower cast microbial contamination using traditional impression and then digital technology; ((**c**), down): epoxy resin cast for full digital prosthesis on implants; ((**d**), down): epoxy resin cast for the onlay fabrication by full technique; ((**c**,**d**), upper): packaging of disinfected personalised medical device (crown and onlay).

**Table 1 sensors-22-00049-t001:** Update of the main differences in cross-infection prevention in the case of traditional technology vs. CAD/CAM technology in dentistry during COVID-19 pandemic [6].

	Need for	Rationale	Traditional Technology	CAD/CAM Technology	Reference
1	Use of all, adequate and certified PPEs (gloves, surgical mask or FFP2 grade mask, shield, gown, cap, impervious body suit) during impression taking	Avoid microbial and SARS-CoV-2 transmission even in vaccinated DHCWs	Yes	Yes	[32,34,45]
2	Follow standard and transmission-based prevention	Some patients are highly susceptible to SARS-CoV-2 infection. No data on the risk of SARS-CoV-2 transmission during dental practice. Low/medium occupational risk from preliminary data.	Yes	Yes	[32,34,45]
3	Attention to OML, chronic sialadenitis, xerostomia, other oral viral infections	Earlier salivary SARS-CoV-2 than lung lesions. Screening of oxygen saturation (cut off < 93%) by non-invasive pulse oximeter.			[23,24,25,26,27,28,29,30,31,35,36,79]
4	Impression material mixing	Time-consuming procedure. Contaminated glovesand transient microbiota on hands.	Yes	No	[6]
5	Attention during impression	After recovery from COVID-19: the respiratory functional recovery (dyspnoea) could be slow and partial. Abdominal breathing (21%).	++++	Easier procedure	[13,30,31,71]
6	Use of sterile steel or plastic impression tray	As the average survival time ofcoronavirus on plastic or steel surfaces is approximately 48–72 h.	Yes	No	[106,107,108]
7	Time for impression procedure	Reduce physical distancing.	++++	++	
8	Good oral hygiene for patients	Reduced oral bioburden.	++++	++	[7,8,12,14,15,16,19,22,35]
9	Pre-procedural mouth rinses (PPMRs) with antimicrobial product	No published evidence regarding their clinical efficacy in reducing SARS-CoV-2 viral load or in preventing transmission. Expected contamination in aerosols and splatter generated during dental procedures and impression contamination.	Yes	Yes	[33,35,45]
10	Laser before impression Proper ventilation Appropriate suction units with in-line filters MIFUs followed for cleaning and disinfection/sterilization of laser, laser pen, and tip	Reduced aerosol and environmental contamination to accommodate for odours, tissue debris, and laser plume suitable for the capture of debris being removed to avoid transmission.	Yes, often for better impression	Yes, not often because of scanner technology	[99]
11	The use of intraoral scanner	Avoid contaminated impression. Patient discomfort, and avoid droplet splatter and aerosol by sneezing and coughing, etc.	Yes, limited with impression scanner	Yes, very often all digital procedures	[6]
12	Impression treatment: early Clean off gross debris and saliva, secured in a plastic leak-proof bag, and sprayed with, or submerged in, a low-level disinfectant following MIFU	Avoid air and environmental contamination. SARS-CoV-2 is a labile infective agent. Spike protein conformation could be influenced by disinfectant pH.	Yes, possible consequence of dimensional stability, impression accuracy, and disinfectant activity	Not needed	[6,105]
13	Dental laboratory prescription Work order	Contaminated paper by SARS-CoV-2, other microbials, blood.	Yes	No, because of e-work order	
14	Clean, package, and decontaminate (if possible sterilise) personalised medical devices before sending for repair or maintenance	To reduce infectious bioburden.	Yes	Yes	[105,110]
15	Use single-use shipping materials (e.g., plastic bags with zip-lock bag and client id) Then packing of the impression without hand contact	To avoid infectious bioburden.	Yes, always for impression and medical devices	Yes, limited only for medical devices	[90,105]
16	Casts with synthetic materials	Not deformable. Disinfection easier.	Not possible or advantageous	Yes	
17	Proper disinfection of all items prior to dispensing or placing in a client’s mouth	Avoid environment, oral, hand infective agents, and work toxic residues.	Yes	Yes	[90,105]
18	Item transport from laboratory in a C/P pouch labelled with the indication “cleaned” plus in a clean, puncture-resistant container for transport	Avoid hand environmental contamination. Avoid infectious agent transmission, including SARS-CoV-2.	Yes	Yes	[105]
19	DUWL maintenance and control of microbial contamination of city water used for dental care following MIFU	Maintain water quality according to national regulation. Reduce air contamination.	Yes, including quality for alginate impression	Yes, but not important for digital impression	[32,45]
20	Ventilation system and air conditioning system	Maintain fresh air or medical grade air. Some patients affected by chronic diseases are particularly susceptible to COVID-19.	Yes	Yes	[32,45]
21	Position of the work station (PC and scanner) for digital impression: near the fresh air flux in relation to air movement in a clean-to-less-clean flow direction;	Limited clinical contact surface contamination.		Yes	As rationally recommended following other instruments [32,45]
22	Routine cleaning and decontamination of the scanner tip, ward, and system, and all other clinical contact surfaces touched during analogic and digital impression	Surface decontamination using registered hospital-grade disinfectants and against SARS-CoV-2; EPA List N.	Yes, for impression material dispenser	Yes	[50,51,52,53,54,55,56,112]
23	Single-use, transparent, and medical-grade barriers	Reduce contamination and decontamination work.	Yes	Yes, important for the scanner tips	[32,45]
24	Scanner tip sterilization	Inactivation of SARS-CoV-2 by steam in class B small autoclave.		Yes	[19,20,21,22,23,24,25,104]
25	Suction lines and HVE	Limited air and clinical contact surface contamination.	Yes	Yes	[32,45]

**Table 2 sensors-22-00049-t002:** Updated characteristics and procedures for IOS reconditioning after the lesson of COVID-19 pandemic [6].

Actions	Current and Future Solutions
Avoid microbial transmission by refraining from touching with bare or contaminated hands or with torn gloves	● Voice commands > Touch screen > Keyboard ● Reduce unnecessary touch/contact points ● Avoid touching the optical surface of the wand ● Remove and replace gloves after each patient procedure ● Fresh gloves before reconditioning (cleaning and disinfection)
	● Medical-grade PC keyboard with low-profile key design and flat keys for easy cleaning and thorough surface disinfection; 360° safeguard against dirt and splashed water; silicone membrane for highest hygiene; sealed key field
	● Medical mouse with a silicone membrane protecting against ingress of dirt and liquids (water, spray liquid disinfectant)
	● Wireless scanner
	● Ease of use
Limit the microbial contamination by reducing contact with respiratory fluids, oral surfaces, saliva splashing, or dirty/deteriorated/scratched mirrors	○ No requirement of contrast medium (powder) ○ Anti-fogging system (with heat > air flux) ○ Scanning speed, accuracy, and precision → scanning time ○ Accuracy of diagonal scanning ○ Prefer smaller wand tip dimension: • Cerec Primescan (10 × 11 mm) • Carestream (13 mm × 13 mm) • 3M true definition (14.4 mm × 16.2 mm) • Planmeca PlanScan (48 mm × 53 mm) • Align iTero element (50 mm × 68 mm) ○ Visually inspect the wand and the scanner tip for signs of deterioration (corrosion, discoloration, pitting, and cracks) or any other kind of damage
Lessen contamination caused by air	✓ Position of the scanner, cart and mobile in relation to dental chair, DHCW, air ventilation and units with HEPA filters
	✓ Disposable sleeves
	✓ Store instrument and accessories in a dry and dust-free location
	✓ Antimicrobial-coated barrier for PC keyboard and mouse
Ensure safe reuse	➢ Clear MIFU and easy reconditioning
− wand	➢ Constructive solutions (smooth and curved surfaces, small-sized joints; removable parts, electric quick couplings) to limit contamination or render it more ergonomic
	➢ Pay attention to the indents, grooves, joints, vents of all parts, and severe kinks in the cable
	➢ Disinfect the first 10 centimetres of sensor cable
	➢ Use compatible cleaners and disinfectants (details in Section 3.4.3)
	➢ Disinfectant on wipes or impregnated wipes for homogeneous dispersion of disinfectant on surfaces that are at all angles, and less dispersion in air, minor vaporization, and inhalation exposure to components
− scanner and scanner tip	➢ Disinfect all the sensors
	➢ Rinse and brush the external surfaces of the tip under warm running water (3–3.5 L/min, 30–35 °C) for a minimum of 15” ➢ Details for cleaners in Section 3.4.3. ➢ Brush tips and scanner ward until visibly clean for at least 30” ➢ Use a soft bristled brush to avoid scratching the mirror ➢ Avoid contact between the mirror and the plastic handle of the bristled brush ➢ Rinse the mirror and surrounding crevice under running water for an additional 15 s ➢ Use a lens tissue or lint-free cloth or inert impregnated wipes to remove anything from the mirror in the tip.
	➢ Autoclavable scanner tips ➢ DO NOT stack tips on or around other metal instruments or near integrator class 5 ➢ Sterilise scanner tips in cassette or wrapped in sterilization pouches/tray
− physical barriers	❖ Hooding of disposable sleeves in front of the patient
	❖ Medical-grade barriers with antimicrobial activity
	❖ Antimicrobial surface of scanner ward
	❖ Optical-medical-grade barriers
	❖ Clean&Remind function with a fading LED, hardware-based, on the keyboards and high-touch surfaces for reminding the user to perform cleaning/disinfecting procedure according to protocol, with a lock in the absence of it

## Data Availability

Not applicable.

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
