# Peer review of "Pros and Cons of CAD/CAM Technology for Infection Prevention in Dental Settings during COVID-19 Outbreak"

_sensors, 2021, doi:10.3390/s22010049_

Round 1

Reviewer 1 Report

The paper addresses the application of CAD/CAM technology in dental practice with the relationship to COVID-19 pandemics. The focus was made on infection prevention.

In general, the topic is interesting. Application of CAD/CAM technology in dentistry is very promising and has already had great results and potential in the future. The major advantages of CAD/CAM technology in dentistry goes far beyond just infection prevention. This is completely out of the scope of the paper. Such a perspective on CAD/CAM in dentistry creates a very limited and hence, a wrong picture.

The problem addressed in the paper is not clearly stated and phrased and therefore it is not clear what actually, is discussed in the paper and problem is analyzed and solved. The paper has poor logical structure and looks like a unstructured collection of facts without proper analysis.

The conclusion of the paper is quite shallow and is limited to some items of minor advice not supported by logical analysis.

Some English phrases in the paper need editing.

In my assessment, the submitted paper cannot be accepted as is and should be rejected. The paper is poorly structured and written. It must be completely rewritten with a clear focus and logical structure, clear problem statement and focused conclusion.

Author Response

Dear Referee,

first of all, we would like to thank you for all suggestions.

Question 1: The major advantages of CAD/CAM technology in dentistry goes far beyond just infection prevention......... This is completely out of the scope of the paper. Such a perspective on CAD/CAM in dentistry creates a very limited and hence, a wrong picture......

A1: The authors know very well that the CAD/CAM technology has several advantages in addition to the infection prevention. This has also been indicated in the previous version. See present version at lines: 39-41,198-202,204-251,266-308. However, this paper is focused on infection prevention and it seems appropriate to underline its importance in the absence of specific guidelines and high-quality MIFU.  We added lines 42-47, and softened lines 205 and 612.

Furthermore, we would like to underline lines 190-195 (still present in the first version). It is true that dental patients like digital technology when choose the dental practitioner and dental office, however, the first criteria is always safety against infection. Dental patients wan SAFETY during dental care with all forms of technology, without exceptions. There is a great deal of evidence since the epidemics of AIDS, Zika virus, Ebola, and now relating to COVID-19.

We added references: 4,5,10,11,17,40,104.

Question 2: The problem addressed in the paper is not clearly stated and phrased and therefore it is not clear what actually, is discussed in the paper and problem is analyzed and solved. The paper has poor logical structure and looks like a unstructured collection of facts without proper analysis.

A2: Only recently has CAD / CAM technology has been used extensively in dentistry and its role in infections prevention is certainly a new item of discussion. The authors believe that this review requires a list of the single items of interest around this topic. This approach is commonly followed by recommendations for emerging technology. Nevertheless, we changed the logical structure of the MS by changing the sections (3.1-3.7). We added lines 400-405.

We  modified subchapters as follows:

The introduction also includes oral mucosal lesions as suggested by another referee

3.1.     Main Advantages of CAD/CAM Technology for COVID-19 Prevention in dentistry

3.1.1           Additional Advantages of digital scanning to avoid compression of mucosal tissues and to detect OML

3.2.             Market Perspective and Dentist Expectations during COVID-19 Pandemic

3.2.1.  Dental CAD/CAM Market Size, Share & Forecast, 2021 – 2027

3.2.2.          Dentist Perspectives

3.3.  The Troubles of Miscellaneous Laws, Guidelines and Recommendations

3.3.1.       Other Recommendations for prosthodontics and their limits

3.3.2.          Recent Indication from the CDC for Environmental Infection Control during the COVID-19 pandemic

3.4.  From recommendations to practice for infection prevention during the COVID-19 pandemic

3.4.1.          Focus on Trouble Regarding Analogic Impression

3.4.2.          SARS-CoV-2 and Factors Influencing Intraoral Scanning and Digital Practice

3.4.3.       Focus on Reconditioning of Digital Scanner and CAD/CAM materials

3.4.3.1.       Reconditioning of Digital Scanner

3.4.3.2.    Current MIFU for Dental Scanner

3.4.4..   Focus on Digital Models

3.5.  The Challenge to Dental Surgery during the COVID-19

3.6.  Medico-Legal Perspective of Providing Dental Care using CAD/CAM technology during the COVID-19 Pandemic

3.7. Future Trend for ICP 

3.7.1.     Reconditioning of CAD/CAM Materials

Question 3: The conclusion of the paper is quite shallow and is limited to some items of minor advice not supported by logical analysis.

A3: This is an opinion not sustained by the current need for patient safety using all forms of technology, including CAD/CAM and, in general, the need to prevent human error and near-misses, in order to avoid adverse events. According to the “Safety first” approach, we added lines 400-405. In the absence of specific recommendation and high-quality MIFU,  the conclusion contains important recommendations for infection prevention and the current approach to patient and occupational safety. Nevertheless, we have cut the last line of the previous conclusion, and modified lines 666-673.

We added ref: 4,5,10,11,17,40,104.

Concerning patient safety in dentistry, we would like to note references 17-29, 31-35, 45 in our recent paper https://www.fortunejournals.com/articles/are-the-guidelines-for-surgical-dental-cares-suitable-for-covid19-pandemic.html

Question 4: English language and style are fine/minor spell check required

A4: it has been done a further revision

Reviewer 2 Report

I would like to give the final assessment about the paper " Pros and cons of CAD/CAM technology for infection prevention in dental settings during COVID-19 outbreak".

Author Response

Dear Referee,

First of all, we would like to thank you for all suggestions.

English language and style are fine/minor spell check required: done further revision

Reviewer 3 Report

I appreciate your work but  I have several recommendations:

  1. I think that you can improve the structuring of the There is a lot of information, but in my opinion there are not structured
  2. In chapter 1. Information Sources and Search Strategy, (row 8-97) please specify clearly the inclusion and exclusion criteria.
  3. In Chapter 3, subchapter 3.1. is part of the introduction rather than the results. ( row 99-193). The title is Pros and cons of CAD/CAM ...  but you only talk about this topic from subchapter 3.2.
  4. Please specify more clearly which are the variables / items you followed in the studied bibliography, as well as the data processing method
  5. The references chapter does not fully respect the recommended style

Author Response

Dear Referee,

first of all, we would like to thank you for all suggestions.

Question 1: I think that you can improve the structuring of the There is a lot of information, but in my opinion there are not structured

A1: CAD / CAM technology has only recently been widely used in dentistry, and its role in infections prevention is a new item of discussion. Authors believe that this review requires a list of the single items of interest around this topic. Nevertheless, we changed the logical structure of the MS by changing the sections (3.1-3.7). We also added and/or modified lines 42-47, 400-405, 666-673. To sustain section 3.5, we added lines 54-59.

Question 2: In chapter 1. Information Sources and Search Strategy, (row 8-97) please specify clearly the inclusion and exclusion criteria.

A2: We modified lines 145-176

Question 3: In Chapter 3, subchapter 3.1. is part of the introduction rather than the results. ( row 99-193). The title is Pros and cons of CAD/CAM ... but you only talk about this topic from subchapter 3.2.

A3: We included previous subchapter 3.1 in the introduction. We are aware that the introduction is very long, but oral mucosal lesion detection will be important for dentistry of the future.

We have modified subchapters as follows:

3.1.     Main Advantages of CAD/CAM Technology for COVID-19 Prevention in dentistry

3.1.1          Additional Advantages of digital scanning to avoid compression of mucosal tissues and to detect OML

3.2.            Market Perspective and Dentist Expectations during COVID-19 Pandemics

3.2.1. Dental CAD/CAM Market Size, Share & Forecast, 2021 – 2027

3.2.2.         Dentist Perspectives

3.3.  The Troubles of Miscellaneous Laws, Guidelines and Recommendations

3.3.1.      Other Recommendations for postodhontics and their limits

3.3.2.         Recent Indication from CDC for Environmental Infection Controls during COVID-19 pandemics

3.4.  From recommendations to practice for infection prevention during COVID-19 pandemic

3.4.1.         Focus on Troubles Regards Analogic Impression

3.4.2.         SARS-CoV-2 and Factors Influencing Intraoral Scanning and Digital Practice

3.4.3.     Focus on Reconditioning of Digital Scanner and CAD/CAM materials

3.4.3.1.     Reconditioning of Digital Scanner

3.4.3.2.  Current MIFU for Dental Scanner

3.4.4..   Focus on Digital Models

3.5.   The Challenge to Dental Surgery during COVID-19

3.6.  Medico-Legal Perspective of Providing Dental Care using CAD/CAM technology during COVID-19 Pandemic

3.7. Future Trend for ICP 

3.7.1.   Reconditioning of CAD/CAM Materials

Question 4: Please specify more clearly which are the variables / items you followed in the studied bibliography, as well as the data processing method

A4: We modified lines 145-176

To clarify the target, we added lines 146-149 and 400-405 and some references on patient safety and occupational safety.

We added references on SARS-CoV-2 receptors on oral mucosal cells, bone cells and osteoblasts to sustain the possible adverse events during oral- implant surgery.

In relation to line1 174-176, the specific knowledge of authors are infection prevention (all authors), oral pathology: (FS, ABG, LB), orthodontia (AB,ADB,UG), and surgery (UG, ABG).

Question 5: The references chapter does not fully respect the recommended style

A5: We checked reference chapter for the recommended style

Round 2

Reviewer 1 Report

1) The review paper should be better structured and logically organized.

2) The conclusions are self-evident, quite shallow, and very generic.

Author Response

Dear Referee,

first of all, we would like to thank you for your comments.

We submit the revised MS (sensors-1472820) following your suggestion.

Question #1: The review paper should be better structured and logically organized

A#1: Text modifications are shown in red. Nevertheless, the review plan was tailored to be interdisciplinary for at least three types of professionals with very different knowledge, training, and skills: dentists, dental nurses, and the infection prevention coordinator. We understand well that you, as dentist or expert of the field, were very bored on some parts and details about problems and errors concerning our specific topic, but we think them not trivial at all and very important for other workers. The review was created to be shared by all dental team, including the occupational physician, to support the risk assessment document and the operative protocols in the absence of specific guidelines. Some  papers (reported below and selected by many regarding patient and occupational safety) sustain our approach.

  • The perceived frequency and impact of adverse events in dentistry. Simon Wright, Cemal Ucer and Stuart Speechley. Faculty Dental Journal January 2018,9,1
  • Systematic review of patient safety interventions in dentistry Edmund Bailey1*, Martin Tickle1 , Stephen Campbell and Lucy O’Malley. BMC Oral Health 2015, 15:152

·     Classifying Adverse Events in the Dental Office. Elsbeth Kalenderian et al. J Patient Saf . 2021,1;17(6):e540-e556.

·     Eleven basic procedures/practices for dental patient safety. Perea Pérez, Bernardo, Labajo González, Elena, Acosta Gío, Enrique A. y Yamalik, Nermin. Journal of Patient Safety, 2015, pp. 1-5.

·     Assessing the Patient Safety Culture in Dentistry A. Yansane et al. JDR Clinical & Translational Res 2020

·     Patient Safety Incidents and Adverse Events in Ambulatory Dental Care: A Systematic Scoping Eduardo Ensaldo-Carrasco et al. J Patient Saf 2016,

·     de Oliveira Corrêa et al. Patient safety in dental care: an integrative review. Cad. Saúde Pública 2020; 36(10):e00197819

Question #2: The conclusions are self-evident, quite swallow and very generic.

A#2: The conclusions have been changed, but what you considered “self-evident, quite swallow and very generic” doesn’t take into account the reality in dentistry regarding the adverse events and errors during infection prevention and related legal consequences.
